

# Width of Plasmaspheric Plumes Related to the Intensity of Geomagnetic Storm

Zhanrong Yang[1,2], Haimeng Li[1,2], Zhigang Yuan[3], Zhihai Ouyang[1,2], Xiaohua Deng[1]

[1] Department of Physics, School of Physics and Materials, Nanchang University, Nanchang 330031, China

[2] Institute of Space Science and Technology, Nanchang University, Nanchang 330031, China

[3] School of Electronic Information, Wuhan University, Wuhan, China

*Correspondence to*: Haimeng Li (lihaimeng@ncu.edu.cn)

## Abstract.

The plume is a plasma region in the magnetosphere that is detached from the main plasmasphere.

It significantly contributes to the dynamic processes in both the inner and outer magnetosphere. In this paper, using Van Allen Probe A, the correlation between plume width and the intensity of a geomagnetic storm is studied. First, through the statistical analysis of all potential plume events, we find that there is almost no correlation between plume width and the intensity of geomagnetic storms. However, for the plumes in the recovery phase after improved sifting, it seems that there

is a negative correlation between the plume width and the absolute value of minimum Dst during a storm. Utilizing test particle simulations, we study the dynamic evolution patterns of plumes during two geomagnetic storms. The simulated structures of the two plasmaspheric plumes are roughly consistent with the structures observed by the Van Allen Probe A. This result suggests that the plasmaspheric particles escape quickly during intense geomagnetic storms, causing the

width of the plume to be relatively narrow during the recovery phase of intense geomagnetic storms. These results are helpful for understanding the dynamic evolution of the plasmasphere and plume during geomagnetic storms.

## 1. Introduction

The plasmasphere is a region of high-density cold particles (at several electron volts) in the inner

magnetosphere. The motions of the outer plasmaspheric particles are periodically driven by geomagnetic activity. During geomagnetic storms, the interplanetary magnetic field (IMF) moves southward and leads to geomagnetic reconnection, which subsequently drives the



convection electric field (Dungey, 1961). Then, plasmaspheric particles move along the E ×B-drift paths in the electric field of the inner magnetosphere and escape from the plasmasphere.

The process is known as plasmaspheric erosion. It will force the plasma to extend sunward and produce plasmaspheric plumes that rotate around the Earth during geomagnetic storm intervals (Lakhina et al., 2000).

Previous studies have indicated that the drift paths of plume plasma are not restricted to the innermost magnetosphere (Spasojevic et al., 2005; 2010). This means that the plasmaspheric

plume is an important channel for the exchange of mass and energy between the inner magnetosphere and outer magnetosphere (Lakhina et al., 2000). Furthermore, although electromagnetic ion cyclotron (EMIC) waves are not preferentially observed in high-density plumes (Usanova et al., 2013; Grison et al., 2018), the plume may be correlated with the excitation of EMIC waves (Grison et al., 2018; Yu et al., 2016; Yuan et al., 2010), and whistler-

mode hiss emissions often exist in plasmaspheric plumes (Su et al., 2018; Kim et al., 2019; Ma et al., 2021). Therefore, understanding the evolution of plumes is essential. When the intensity of geomagnetic storms increases, plasmaspheric erosion becomes stronger with the enhancement of the convection electric field (Chen and Grebowsky, 1974; Grebowsky, 1970). However, relatively less research has been conducted regarding the shapes of plasmaspheric plumes.

Borovsky et al. (2008) statistically calculated the linear relationship between the width of the plasmaspheric plume and the intensity of geomagnetic storms. Borovsky et al. (2008) suggested that the linear correlation coefficient between them was almost 0. Since the plasmasphere can be eroded by the enhanced convection electric field during geomagnetic storms (Krall et al., 2017), the enhanced convection field causes low energy plasma drainage to the magnetopause (Denton

et al., 2005). We consider that the level of storm intensity may affect the width of the plume in some conditions.

In this paper, we utilized the data recorded by Van Allen Probe A (from 2013 to 2018) to identify plasmaspheric plumes. The correlation coefficient between the width of plasmaspheric plumes and the intensity of geomagnetic storms was calculated under different standards.

Furthermore, we ran group test particle simulations to support the statistical results.



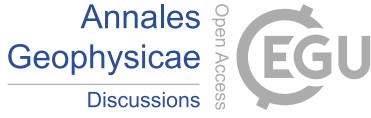

## 2. Data and Statistical Widths of Plasmaspheric Plumes

The Van Allen Probe A (VAP-A) spacecraft was in a highly elliptical (1.1 × 5.8 $R_E$), low-inclination (10°) orbit (Mauk et al., 2013) and collected data from August 2012 to October 2019. We use the electron density data provided by the VAP-A (listed in Level-4 data sets), which

were calculated from the upper hybrid resonance frequency (Kurth et al., 2015). In this study, by analyzing the electron density measurements of VAP-A, the structure of plasmaspheric plumes is determined by the two following criteria: (1) The location of the plasmapause is considered to be the position where the electron density decreases to less than 0.2 times within the 0.5 $R_E$. (2) According to the method commonly used in previous studies, if VAP-A is located outside the

location of the plasmapause and the detected density exceeds the result of the plasmaspheric density model given by Sheeley et al. (2001) (the formula is shown below) for more than 10 minutes, we consider it a plasmaspheric plume (Moldwin et al., 2002; Zhang et al., 2019).

$$n_e = 1390(\frac{3}{L})^{4.83} - 240(\frac{3}{L})^{3.60} \tag{1}$$

A typical example of a plume is exhibited in Figure 1. The blue curve indicates the observed electron density from 08:00 UT to 16:36 UT on 09 October 2016. The red curve indicates the

empirical electron density obtained from the plasmaspheric model published in Sheeley et al. (2001). The plasmapause positions are marked by vertical black lines based on the criterion above. In addition, a typical plasmaspheric plume is identified and marked by a gray shadow in Figure 1.

As Li et al. (2022) suggested, the number of plume events in the initial and main phases of

geomagnetic storms is very small, and most plumes mainly form in the recovery phase. Partial plume events can still be observed when the geomagnetic activity recovers to quiet conditions after geomagnetic storms because a relatively long time is required for the plasmasphere to recover to normal levels. Besides, referring to Halford et al. (2010) and Wang et al. (2016), the onset time of storm is defined as the time when the Dst index slope becomes negative and keeps

relatively negative till the minimum of Dst index in our study. In this paper, first we focus on the relationship between the width of the plume in the interval of 10 days after the storm minimum Dst and the corresponding intensity of the geomagnetic storm (represented by the minimum Dst). According to the plume determination criteria above, we find 423 orbits with plumes (within 10





days after the minimum Dst of storm) by searching 4030 VAP-A orbits from 2013 to 2018. Since
several plumes may be identified in some orbits, a total number of 586 plume events are found.

The concept of the observed plume width only has a 'fuzzy' definition in the literature. To ensure
the accuracy of the analysis, two judgments are adopted to represent the detected plume width.
First, the width of the detected plume is defined as the Cartesian distance ($\Delta P$) between the two
ends (the entrance and exit) of the VAP-A orbit where the plume is detected:

$$\Delta P = \sqrt{[(X_0 - X_1)^2 + (Y_0 - Y_1)^2 + (Z_0 - Z_1)^2]} \qquad (2)$$

where $X_0, Y_0, Z_0$ represent Cartesain position of plume entry edge, and $X_1, Y_1, Z_1$ represent
Cartesain position of plume exit edge.

Second, the width is considered the difference between the MLTs of the two plume ends
($\Delta MLT$). In this study, the corresponding geomagnetic index of the plasmaspheric plume is
considered to be the minimum Dst value in the geomagnetic storm. In addition, a complication in
measuring the plume width is that the plume is still rotating when VAP-A passes through the
plasmasphere, which will lead to more or fewer satellite orbits in the plasmasphere. For
statistical significance, the average width can reduce this influence because of the similar
behavior between the entry and exit edges of the plume, and it also reminds the reader that the
width measurement of the individual plume may be some error (Borovsky et al., 2008). To
clearly reflect the statistical variation in plume width associated with geomagnetic activity, we
use the averaged width of the detected plume in steps of 5 nT (Dst index) to represent the plume
width in the corresponding Dst range in the study.

Figure 2 shows the widths of the 586 plumes detected above as a function of the corresponding
Dst. The Cartesian distance ($\Delta P$) and $\Delta MLTs$ are denoted by the black solid points in Figure 2a
and 2b, respectively. The red curves connect the averaged widths of plasmaspheric plumes in
each step of the 5 nT range (plotted by red asterisks). Then, we fit the red asterisk dots through a
linear function, which is drawn by a blue line. It is obvious that the blue curves are almost
parallel to the Dst indices for both Figures 2a and 2b. This means that the plume width is
independent of the intensity of the geomagnetic storm. For Figure 2a, the calculated Pearson
correlation coefficient (marked as R), which indicates the relevance between the averaged
Cartesian distance of plasmaspheric plumes (indicated by red asterisks) and the corresponding
Dst value, is only -0.017137. The Spearman correlation coefficient (marked as $\rho$), which is



generally adopted to express the reliability of their linear correlation, reaches 0.9352. For Figure 2b, the calculated Pearson correlation coefficient between the averaged ΔMLT of plasmaspheric

plumes and the corresponding Dst value is only -0.052509. The Spearman correlation coefficient is 0.80315. The low R and high $\rho$ values in Figures 2a and 2b suggest that there is almost no correlation between the widths of the above plumes and the corresponding Dst value, and their relationship may be very complicated.

The formula of the Pearson correlation coefficient is:

$$R = \frac{N \sum_i x_i y_i - \sum_i x_i \sum_i y_i}{\sqrt{N \sum_i x_i^2 - (\sum_i x_i)^2}\sqrt{N \sum_i y_i^2 - (\sum_i y_i)^2}} \tag{3}$$

The formula of the Spearman correlation coefficient is:

$$\rho = \frac{\sum_i (x_i - \bar{x})(y_i - \bar{y})}{\sqrt{\sum_i (x_i - \bar{x})^2 \sum_i (y_i - \bar{y})^2}} \tag{4}$$

where $x_i$ and $y_i$ are two sets of data with the same number (N).

An explanation of the poor relevance is that some processes may influence the structure of the plasmasphere and plume. For example, the advent of quiet conditions after geomagnetic storms can contribute to the refilling of the plasmasphere because ionospheric particles are drawn

upward from low altitudes along magnetic field lines. In addition, the number of samples with geomagnetic storms that are too strong or too weak may be too few to have statistical significance. Therefore, to better understand the influence of storm intensity on the plume width, we further sifted the plume events.



First, considering that the plasmasphere can be obviously refilled after the time of geomagnetic
disturbance, we only retain the events during the recovery phase of the geomagnetic storm. This
operation ensures that the main factor affecting the structure of the plume is the erosion process
of the geomagnetic storm, which is the main topic in the study. Similar to the standards described
in Engebretson et al. (2008), Halford et al. (2010) and Bortnik et al. (2008), we define the end of
the recovery phase as 5 days after the main phase finishes. The numbers of plume events
corresponding to different time intervals (here, 1 day represents the interval of 0-24 hours) after
the minimum Dst are shown in Table 1. The total number of plume events in the recovery phase
(the interval time is 1-5 days) is 377.

Among the 377 plume events, the minimum Dst value of the intensest geomagnetic storm
reaches -209 nT, but 68% of plume events (256 events) correspond to Dst values ranging from -
70 nT to -15 nT, and 88% of plume events (333 events) correspond tthe minimum Dst value of
the intensest geomagnetic storm ro Dst values ranging from -90 nT to -15 nT. For the accuracy
of statistical research, we exclude extremely intense storms in the study, and we only statistically
analyze plasmaspheric plume events from -70 nT to -15 nT (and from -90 nT to -15 nT) during
the recovery phase.

Moreover, in addition to the density exceeding the Sheeley model for more than 10 minutes
outside the plasmapause, we further improve the standard of plume judgment. Referencing the
method of Darrouzet et al. (2008), the $\Delta L$ of the structure (the difference in the L shell between
the entrance and exit of the plume orbit) should be large enough (0.2) to be considered a plume.
On the other hand, the events with excessive linear width ($>3.5$ $R_E$) are also considered not
plumes because they are more likely to be the cross section of the plasmasphere rather than the
plume. Based on this standard, we exclude 111 plume events in the range from -70 nT to -15 nT
and 168 plume events in the range from -90 nT to -15 nT.

The process of deleting events that do not match the standard is shown in Figure 3. The orange
solid dots indicate the events that are not in the recovery phase, which are beyond 5 days after
the minimum Dst. The purple solid dots indicate plume events with corresponding Dst values
less than -70 nT or greater than -15 nT. The blue and red solid dots represent the tracks with $\Delta L$
less than 0.2 and Cartesian distances greater than 3.5 $R_E$, respectively. The gray and black solid
dots indicate the retained plume events with corresponding Dst ranges from -90 nT to -70 nT





and from -70 nT to -15 nT, respectively. Ultimately, there are 145 retained plume events in the
Dst range from -70 nT to -15 nT, and 165 retained plume events in the Dst range from -90 nT to
-15 nT. In addition, the spatial distribution of 586 events is shown in Figure 4. The curves
represent the VAP-A orbits while the events are observed. The colors of the curves represent the
filtering process discussed above, and the corresponding event color is consistent with Figure 3.
It also shows that the retained plumes (indicated by the black curves) are mainly observed on the
dusk side (MLT~ 15:00 to ~21:00).

Figures 5a and 5b show the correlation analysis between the retained plume width and storm
intensity with a minimum Dst from -70 nT to -15 nT. The formats are similar to Figure 2a and 2b.
Completely different from the results before the sifting plume events (as shown in Figure 2), as
indicated by the blue lines in Figure 5, there is a considerable negative correlation between the
plume width and corresponding storm intensity. This implies that as the minimum Dst value
becomes lower, the width of the plasmaspheric plume tends to become narrower. As presented in
Figure 5a, the Pearson correlation coefficient between the averaged Cartesian distance of plumes
and the Dst value reaches 0.61934. The value of the Spearman correlation coefficient is
0.042149, and the low value of the Spearman correlation coefficient means that it is feasible to
express the relationship as a linear one. As presented in Figure 5b, the Pearson correlation
coefficient between the averaged ΔMLT and the Dst value reaches 0.54631, and the Spearman
correlation coefficient is 0.067833. Both the interpretations from Figure 5a and 5b imply that
there is a roughly negative correlation between the width of the plume in the recovery phase and
the intensity of the geomagnetic storm.

Similarly, Figures 5c and 5d show the correlation analysis between the retained plume width and
storm intensity with a minimum Dst from -90 nT to -15 nT. As exhibited in Figure 5c, the
Pearson correlation coefficient between the averaged cartesian distance of plumes and the Dst
value is 0.58. The value of the Spearman correlation coefficient is 0.02131. This also implies that
there is a roughly negative correlation between the plume width and intensity of geomagnetic
storms, with a minimum Dst from -90 nT to -15 nT. As presented in Figure 5d, the Pearson
correlation coefficient between the averaged ΔMLT and the Dst value is 0.37, and the Spearman
correlation coefficient is 0.17. From the perspective of ΔMLT analysis, it seems that the negative
correlation in the Dst range from -90 nT to -15 nT is weaker.



For the accuracy of statistical research, we exclude extremely intense storms in the study, the plume events correspond to intervals of minimum Dst indices from -70 nT to -15 nT and -90 nT to -15 nT are statistically analyzed, respectively.

It seems that the negative correlation for the events with minimum Dst from -90 nT to -15 nT decreases slightly compared to those with minimum Dst from -70 nT to -15 nT.

In this study, the Cartesian distance and ΔMLT are adopted to represent the detected plume width. Both methods imply that there is a negative correlation between the width of the plume in the recovery phase and the intensity of the geomagnetic storm. To compare the similarities and differences between the two standards, Figure 6 exhibits the relationship between $\Delta L/<L>$ and the ΔMLT of plumes, where ΔL indicates the difference between the L shells of the entrance and exit of the plume detected by VAP-A, and $<L>$ indicates the average value of L on the plume orbit. There is a positive Pearson correlation coefficient (0.47109 for the Dst range from -70 nT to -15 nT, 0.39638 for the Dst range from -90 nT to -15 nT) between them, which means that when $\Delta L/<L>$ increases, the tendency of ΔMLT also enhances, although this positive correlation is not too strong. The minimal Spearman correlation coefficient ($\sim 2.89 \times 10^{-10}$ and $\sim 4.30 \times 10^{-7}$, respectively) also shows that the linear relationship between them is very significant.

## 3. Simulation of Plume Evolution

To clearly exhibit the effect of the storm intensity on the width of the plasmaspheric plume, the evolutions of the plumes during two geomagnetic storms (with minimum Dst values equal to -39 nT and -74 nT) are simulated through test particle simulations. In this study, this process differs from the PTP simulation in Goldstein et al. (2004; 2014a; 2014b); the test particle simulations in this study also calculate the evolution of density in the plasmasphere and plasmaspheric plume.

### 3.1. Model Inputs

This simulation assumes that all particles move in the dipole magnetic field model. Considering not only that the plasma motion during the geomagnetic storm will be driven by the combined action of the corotating electric field and convection electric field but also that the subauroral



polarization stream (SAPS) will play a significant modification of the convection electric field, the magnetospheric electric model is assumed to consist of three parts as follows:

[1] the corotation electric potential $\Phi_{rot}$, whose formula is:

$$\Phi_{rot} = -\, C\, \frac{R_E}{r} \qquad (5)$$

where C indicates a constant of 92, which is provided by Völk (1970), and r indicates the geocentric latitude.

[2] the convection electric potential, which is expressed as:

$$\Phi_{VS} = -\, E_{IM} r^2 \sin\varphi\, (6.6 R_E)^{-1} \qquad (6)$$

where $E_{IM}$ is the inner magnetospheric electric field. While the southward IMF turns southward, $E_{IM} = 0.12 \cdot |E_{SW}|$, and when the IMF is reverse, it is equal to $0.12 \cdot 0.25\ \mathrm{mV\,m^{-1}}$. Here, the $E_{SW}$ is the solar wind electric field (Maynard and Chen, 1975). $\varphi$ indicates the azimuthal angle.

[3] SAPS electric potential is an intense, radially narrow, westward flow channel that is mainly located in the dusk-to-midnight MLT area (Burke et al., 1998; Foster et al., 2002) and is considered to significantly modify convection. As Goldstein et al. (2005a; 2005b; 2014b) suggest, the effects of the SAPS in the equatorial magnetosphere are driven by the Kp index, and the electric potential is described as follows:

$$\Phi_s(r, \varphi, t) = -\, F(r, \varphi) G(\varphi) V_S(t) \qquad (4)$$

where $F(r, \varphi)$, $G(\varphi)$ and $V_S(t)$ are functions parameterized by the magnetic latitude, MLT, and Kp index.

The initial plasma density distribution is assumed to change as a function of the L shell ($2 \le L \le 7$) according to the model obtained from Sheeley et al. (2001) (the formula of the Sheeley Model is expressed as Eq 1.) with no MLT dependence. In addition, for regions where the L shell is larger than 7, the electron density remains at 5 cm$^{-3}$. All particles (approximately 128,000 electrons in total) emitted in the model are considered to be cold electrons and assumed to have an initial energy of 10 eV. Here, the motions of electrons are assumed to be adiabatic. We calculate the drift velocity as a combination of the velocity due to $E \times B$ drift and the bounce-averaged velocity due to gradient and curvature drifts (Roederer, 1970; Ganushkina et al., 2005;



Li et al., 2021). Here, the pitch angles of the 128,000 electrons are deemed arbitrary, because the
electron energy is considered to be small enough that the associated gradient-curvature drift
velocity is very small (Roederer and Zhang, 2016). The motions of electrons are mainly
contributed by the $E \times B$ drift. According to the simulation results, we can calculate the particle
density in a certain area to reflect the evolution of the plasmasphere and plume. Notably, the
actual shape of the plasmapause is too complicated to obtain its actual electron distribution
function, so the above typical model electron density distribution is adopted in the research.

### 3.2. Evolution of Plume from 19 to 20 May 2017

The geomagnetic and solar wind indices during the geomagnetic storm from 19 to 20 May 2017
are displayed in Figure 7. As shown in Figure 7b, at 07:02 UT, the IMF turned southward,
leading to a larger negative $B_Z$, and the geomagnetic storm began. The beginning of the
geomagnetic storm is denoted by the blue vertical dashed line. The minimum Dst index of this
geomagnetic storm was -39 nT. As shown in Figure 7c, during the main phase of this
geomagnetic storm (from approximately 12:00 UT on 19 May to 09:00 UT on 20 May), the
maximum EIM index reached 0.4684 mV/m. As shown in Figure 7d, the Kp index that drove the
SAPS model reached a maximum of 4+.

The test particle simulation started at 07:02 UT on 19 May 2017 (the beginning of the
geomagnetic storm). The initial distribution of particle density is shown in Figure 8a. In the
contribution of the convection electric field, the particles in the plasmasphere obviously move
sunward within 5 hours from 07:02 UT to 12:02 UT on 19 May 2017 (as shown in Figure 8b).
Meanwhile, some of the plasmaspheric particles expand to high locations with L>8 and may be
lost to the magnetopause boundary (Spasojevic et al., 2005). During the interval of the next five
hours, more particles move sunward and reach the model boundary (as shown in Figure 8c), and
the L shell of the plasmapause on the nightside obviously decreases. As shown in Figure 8d and
8e, from 22:02 UT to 03:02 UT on 20 May 2017, the width of the plasmaspheric bulge gradually
shrinks, and a plume gradually forms on the afternoon side. Furthermore, under the action of a
corotation electric field, the plasmaspheric plume gradually shifts toward the nightside. From
08:02 UT to 13:02 UT on 20 May (as shown in Figure 8f and 8g), as the convection electric field
and SAPS electric field increase again, a large number of particles move in the sunward direction,



and the plasmaspheric plume narrows with time. For most times in the recovery phase, the convection electric field becomes weaker, and the formed plasmaspheric plume slowly rotates from the afternoon side to the nightside (as shown in Figure 8g to 8h). The plasma density detected by VAP-A from 17:36 UT to 22:39 UT on 20 May 2017 is shown in Figure 8i. From 18:12 UT to 19:53 UT on 20 May 2017, VAP-A operated near the apogee of its orbit on the dusk side, and the plume also rotated to the dusk side. The orbit of VAP-A while it actually observed the plume is indicated by the black curve in Figure 8h. We can see that the observed plume roughly coincides with the simulated plume in Figure 8h. As the positions of the simulated plume and observed plume at this time are roughly identical, we believe that the initial distribution of particles and magnetospheric electric field models used in this paper are basically reliable.

### 3.3. Evolution of Plume from 8-10 June 2015

The geomagnetic and solar wind indices during the geomagnetic storm from 08 to 10 June 2015 are shown in Figure 9. The geomagnetic storm began at 00:18 UT on 08 June 2015 (denoted by the blue line). As shown in Figure 9a, the minimum Dst index of this geomagnetic storm was -74 nT, which was much lower than that during the storm presented in section 3.2. At the beginning of the geomagnetic storm, the $B_Z$ turned southward, and the $E_{IM}$ value (calculated from $E_{SW}$) became a relatively large positive value. The maximum $E_{IM}$ in this geomagnetic storm was 1.0674 mV/m, which was much greater than the 0.4684 mV/m obtained from the geomagnetic storm presented in section 3.2. Meanwhile, the maximum Kp index reached 6 in the main phase. It was also much larger than 4+ presented in the last storm. Both larger $E_{IM}$ and Kp indices implied that convection during this geomagnetic storm was much more intense than that during the geomagnetic storm from 19 to 20 May 2017.

Figure 10a shows the electron density distribution during the first minute of the simulation during this geomagnetic storm. Then, under the contribution of a continuous convection electric field, the plasma in the outer part of the plasmasphere moves along the E×B-drift paths from 00:19 UT on 08 June to 06:39 UT on 10 June. As shown in Figure 10b and 10c, the intenser convection electric field brings about a larger number of particles on the dayside moving sunward and extending out of the model boundary, and the particles on the nightside in the



plasmasphere move faster toward the Earth. As shown in Figure 10d, the particles in the outer part of the plasmasphere dissipate at 00:18 UT on 09 June, and a narrower plume emerges near the dusk side. During the recovery phase of the geomagnetic storm, as shown from Figure 10e to

10h, the formed narrow plume revolves around the Earth. Finally, VAP-A observes the structure of the plume from 06:39 UT to 07:31 UT on 10 June 2015 (as shown in Figure 10i), and the orbit of the probe during this time interval is indicated in Figure h. Both the observations and simulations suggest that the L shell of the plasmapause is lower than that presented in section 3.2. Although there is some difference in the MLT between the simulated plume and observed plume

(approximately 1 MLT, which may be due to a simulation time that is too long), the results imply that the width of the plume is much narrower than that during the geomagnetic storm from 19 to 20 May 2017.

### 3.4. Comparison of Simulation Results

To better exhibit the difference in the simulated plume width during the above two geomagnetic

storms, we calculated the Cartesian distances and ΔMLTs at positions where the L shell was equal to 5, 5.5 and 6. The calculated results are shown in Table 2. As presented in Table 2, at the same L shell, regardless of cartesian distances and ΔMLT, the simulated plume driven by stronger geomagnetic storms is always narrower.

### 4. Discussion and Conclusion

In this study, we present statistical research on the relationship between the widths of plumes and the intensities of geomagnetic storms by analyzing the data collected by the VAP-A. Here, the widths of the detected plume are defined as the Cartesian distance and ΔMLT of the detected plume orbit. In the first step, by directly analyzing all 586 potential plume events after the minimum Dst of a geomagnetic storm, we find that there is almost no correlation between plume

width and the intensity of the storm. This result is similar to the conclusion obtained from Borovsky et al. (2008), which suggests that the linear correlation coefficient between them is almost 0. Since the plasmasphere can be eroded by the enhanced convection electric field during geomagnetic storms (Krall et al., 2017), the enhanced convection field causes low energy plasma drainage to the magnetopause (Denton et al., 2005). We consider that the level of storm intensity



may affect the width of the plume in some conditions. In the second step, we define the end of
      the recovery phase as 5 days after the main phase finishes. Only the plumes in the recovery phase
      interval are analyzed. Moreover, the criterion of plumes for statistical investigation is further
      improved. This result suggests that there is a negative correlation between the plume width and
      absolute value of the minimum Dst value during the storm, although the negative correlation is
not very strong.

      To explain the negative correlation between them during the recovery phase of the geomagnetic
      storm, the group test particle simulation is adopted to reveal the dynamic evolutions of the
      plasmasphere and plume during two geomagnetic storms (with minimum Dst values of -39 nT
      and -74 nT, respectively). By comparing the evolutions during the two storms, we find that in the
more intense geomagnetic storm, the erosion of the plasmasphere is more severe, most particles
      in the outer part of the plasmasphere are dissipated during the initial and main phases, and a
      narrower plume is exhibited during the recovery phase. Although the evolutions of
      plasmaspheres and plumes may be very complicated, the above simulation results provide a
      reasonable candidate explanation for the negative correlation between storm intensity and plume
width during the recovery phase of geomagnetic storms.javascript:void(0);

      As shown in Figure 4, most plume events are mainly observed on the dusk side, the relationship
      between the plume width and its MLT is also a meaningful work, which will be studied in our
      next project. Since there maybe a short time delays with several minutes between the changes of
      $E_{SW}$ and $E_{IM}$ (Nishimura et al., 2009), a more precise of real time $E_{IM}$ model will be discussed
and explored in the future.

      **Data availability**

      The data of EMFISIS aboard the Van Allen Probes are publicly available at the EMFISIS
      website (http:// emfifisis.physics.uiowa.edu/Flight/, last access: 10 January 2022, Kletzing et al.,
      2013). The OMNI data are provided at SPDF website (http://cdaweb.gsfc.nasa.gov, last access:
10 January 2022, NASA, 2022).



**Author contributions**

The conceptional idea of this study was developed by HL and ZY. HL and ZY wrote the paper, and ZY revised it. ZO and XD helped substantially with the analysis. ZY and ZO contributed to the Van Allen Probe data processing. All authors discussed the results.

**Competing interests.**

The contact author has declared that neither they nor their co-authors have any competing interests.

**Acknowledgments**

This research is supported by the National Natural Science Foundation of China (Nos.
42064009). The data of EMFISIS onboard Van Allen Probe are from (http://emfifisis.physics.uiowa.edu/Flight/. The Dst and KP data are provided by OMNI at http://cdaweb.gsfc.nasa.gov

**Financial support.**

This research has been supported by the National Natural Science Foundation of China (grant
nos. 42064009).



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



**Figure 1.**

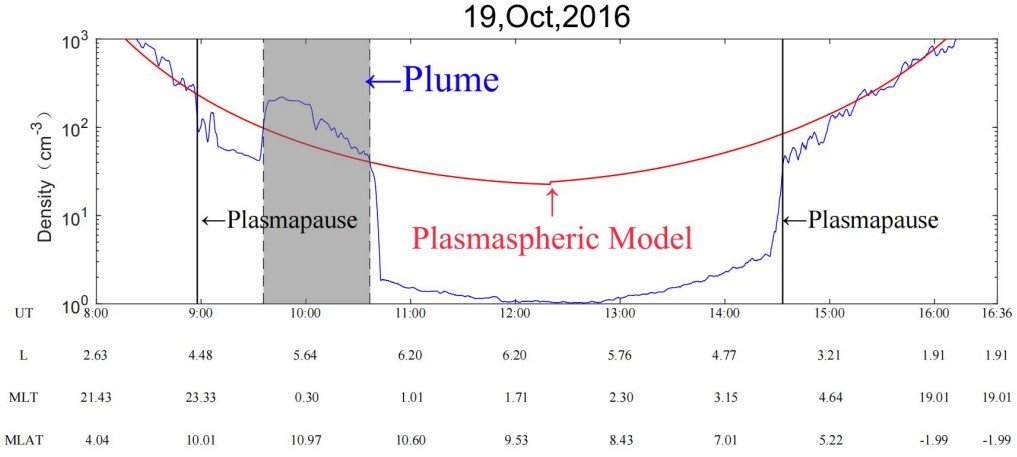

490 **Figure 1.** A typical example of a plume on 19 October 2016. The blue curve represents the detected plasma density, and the red curve displays the density calculated by Sheeley et al. (2001). The position of the plasmapause is marked by black vertical lines. The gray shadow indicates the plasmaspheric plume detected by VAP-A.



**Figure 2.**

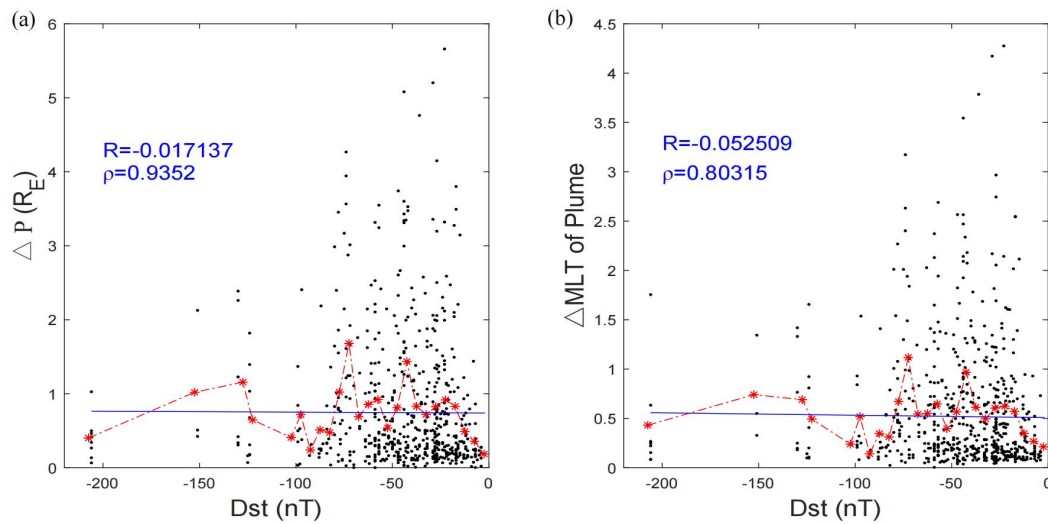


**Figure 2.** (a) The widths of plumes as a function of Dst values are represented by black solid dots, where the width expresses the Cartesian distance between the two ends (the entrance and exit) of the VAP-A orbit where the plume is detected. The linear fitting of the plasmaspheric plume averaged widths in each step of the 5 nT range (red asterisk points) is indicated by the blue line. (b) The format is similar to (a); however, the ΔMLT of the plasmaspheric plume is adopted to represent the width of the plume.

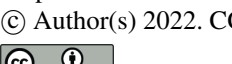



**Figure 3.**

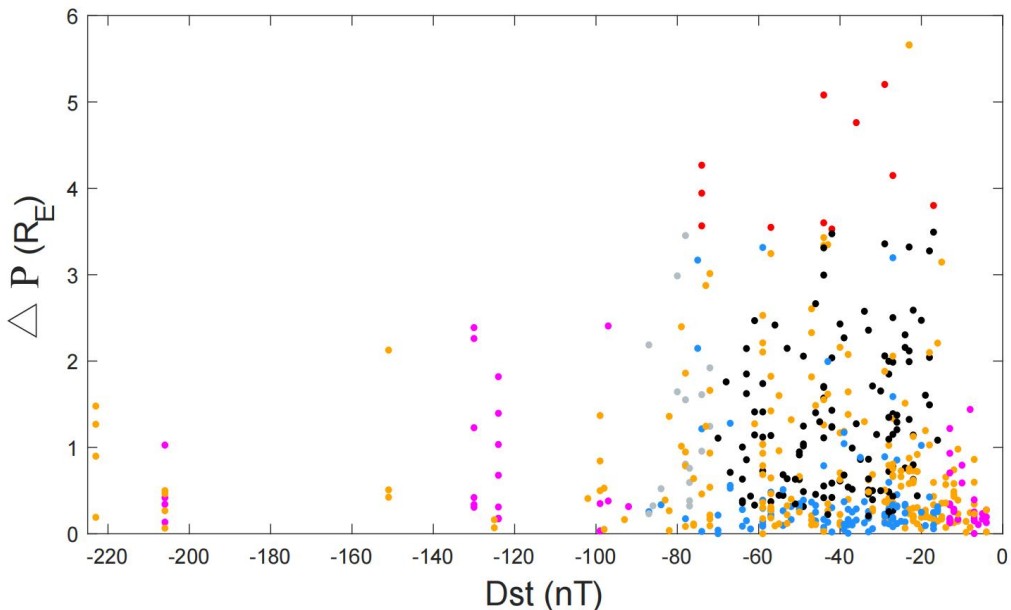

**Figure 3.** The category of observed events: the orange solid dots indicate plume events that are
not in the recovery phase. The purple solid dots display the events with a corresponding Dst
index less than -70 nT or greater than -15 nT. The blue and red solid dots represent the events
with $\Delta$L less than 0.2 and Cartesian distances greater than 3.5 $R_E$, respectively. The events in the
Dst range of -70 nT to -15 nT and -90 nT to -70 nT eventually retained are indicated by black
and gray dots, respectively.





**Figure 4.**

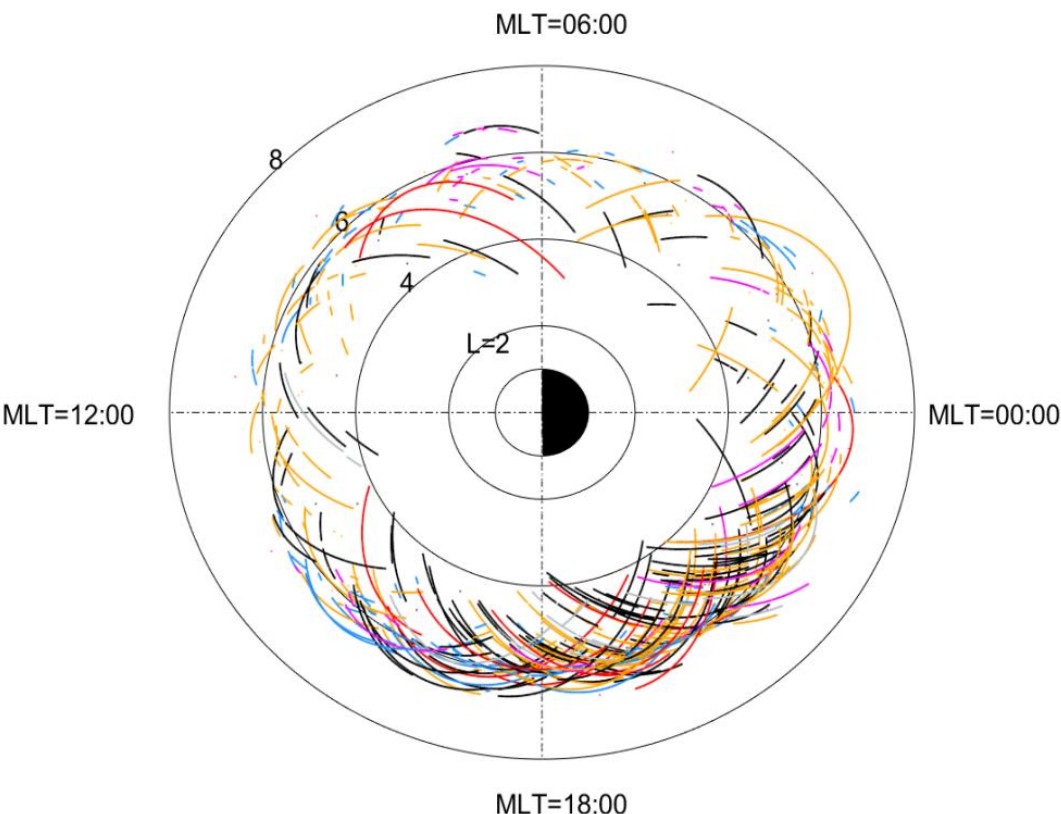

**Figure 4.** The spatial distribution of 586 plasmaspheric plumes is shown in the MLT‑L plane.

The color codes are the same as those in Figure 3.



**Figure 5.**

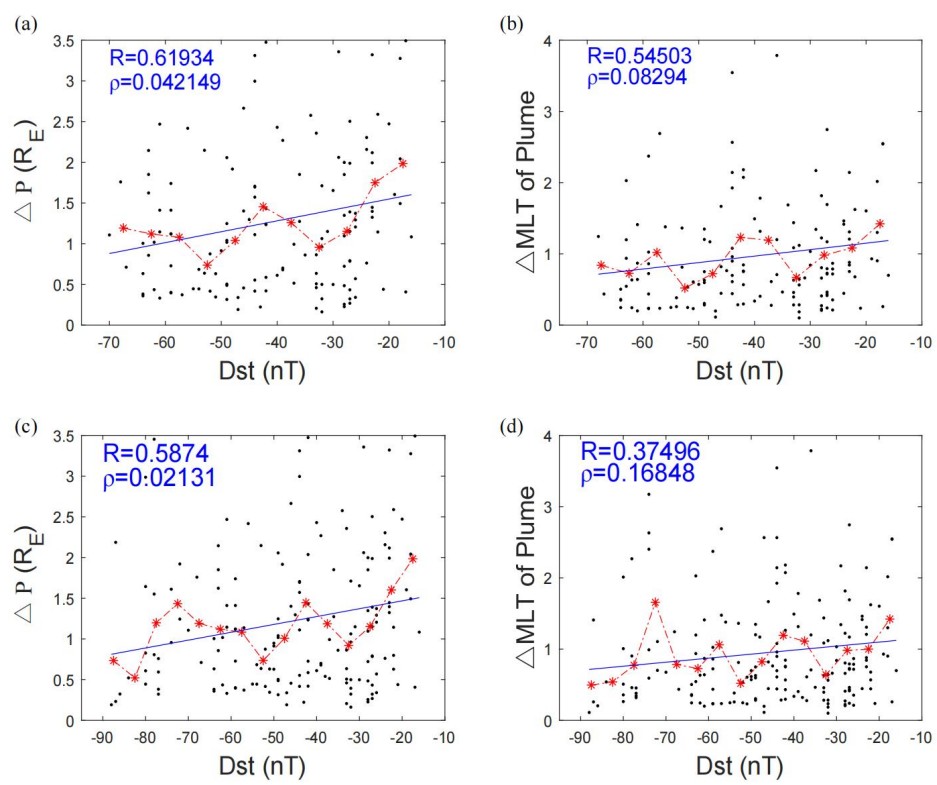

**Figure 5.** The format is the same as in Figure 2. However, only the retained plume events that meet more stringent sifting conditions are analyzed.

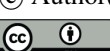



**Figure 6.**

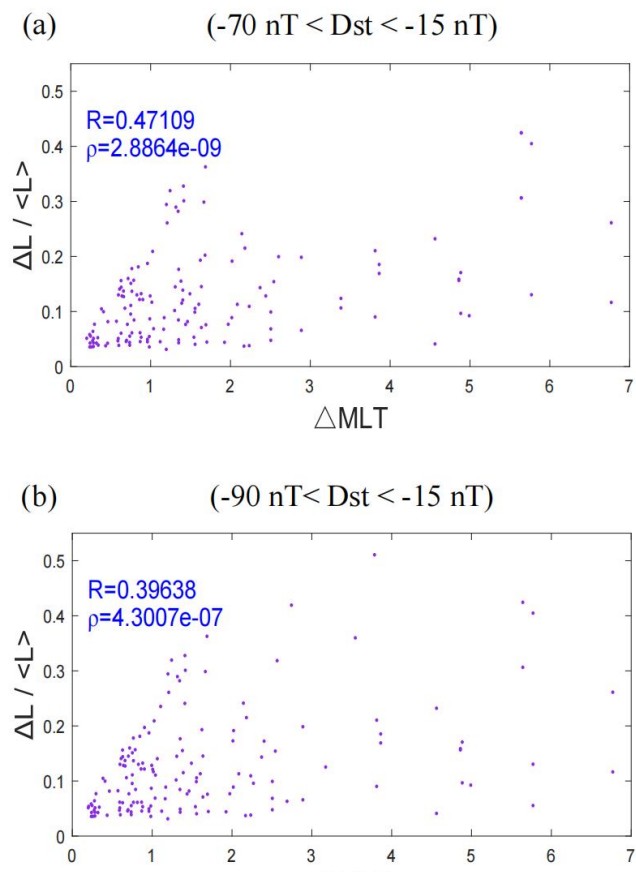

**Figure 6.** The relationship between ΔL/<L> and ΔMLT.





**Figure 7.**

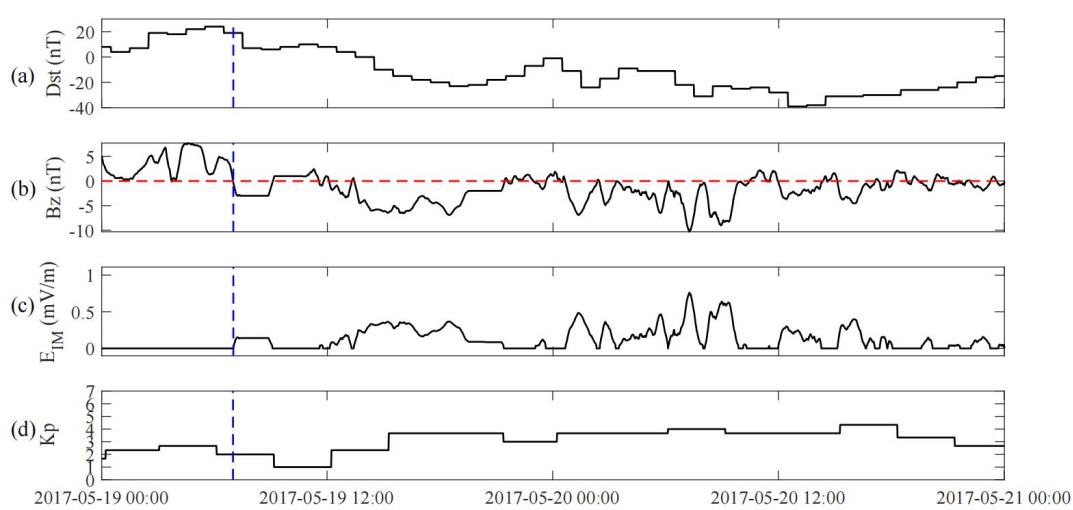

**Figure 7.** The indices of geomagnetic activity and solar wind during the geomagnetic storm that occurred in the time interval of 19-20 May 2017: (a) Dst index, (b) $B_Z$ index in GSM coordinates. The red dotted line indicates the position where $B_Z$ is equal to 0, (c) $E_{IM}$ index and (d) Kp index. The start time of the geomagnetic storm (07:02 UT on 19 May 2017) is marked by the blue vertical dashed line.

**Figure 8.**

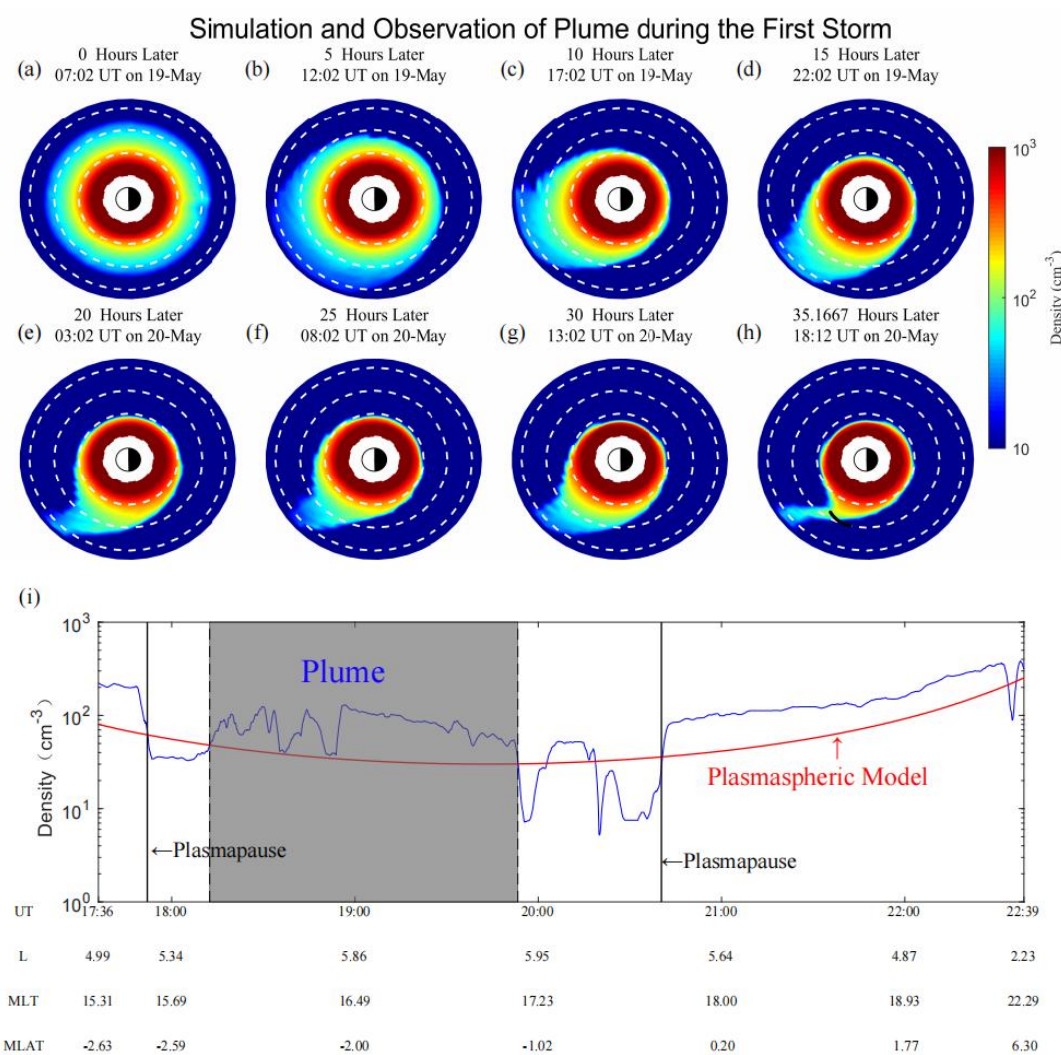

**Figure 8.** (a-h) Equatorial plots of the plasmasphere and plume obtained by simulation in the
time interval of 19-20 May 2017. The white dotted circles, from inside out, indicate L shell
values of 4, 6, and 8. The simulation duration and the corresponding actual duration are
represented in the title. The solid black line in Figure 8h represents the orbit arc in which VAP-A
observed a plasmapheric plume from 18:12 UT to 19:53 UT on 20 May. (i) The plasma density
detected by VAP-A from 17:36 UT to 22:39 UT on 20 May 2017. The gray shadow indicates the
plasmaspheric plume detected by VAP-A.



**Figure 9.**

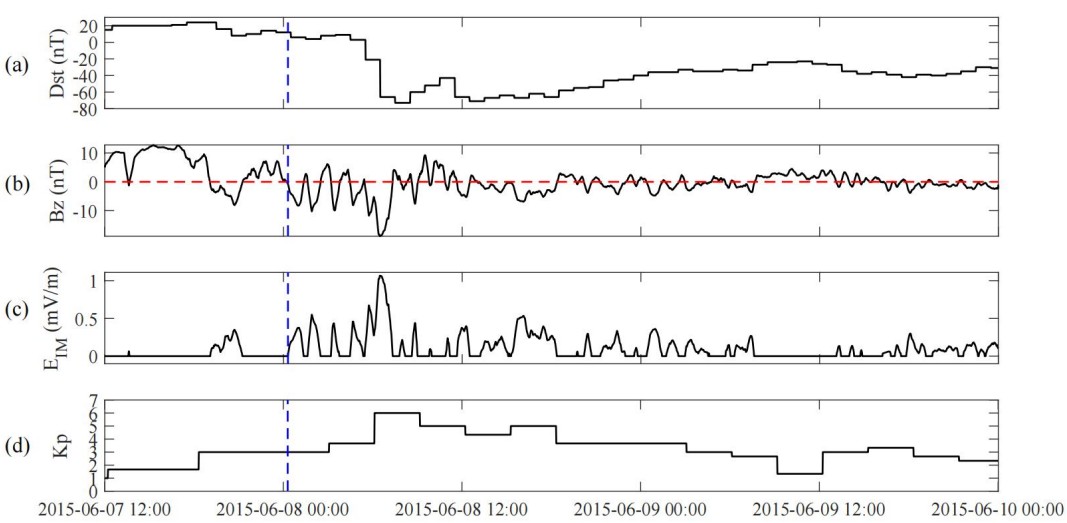

**Figure 9.** The indices of geomagnetic activity and solar wind during the geomagnetic storm that occurred in the time interval of 08-10 June 2015; the format is the same as that in Figure 7.

**Figure 10.**

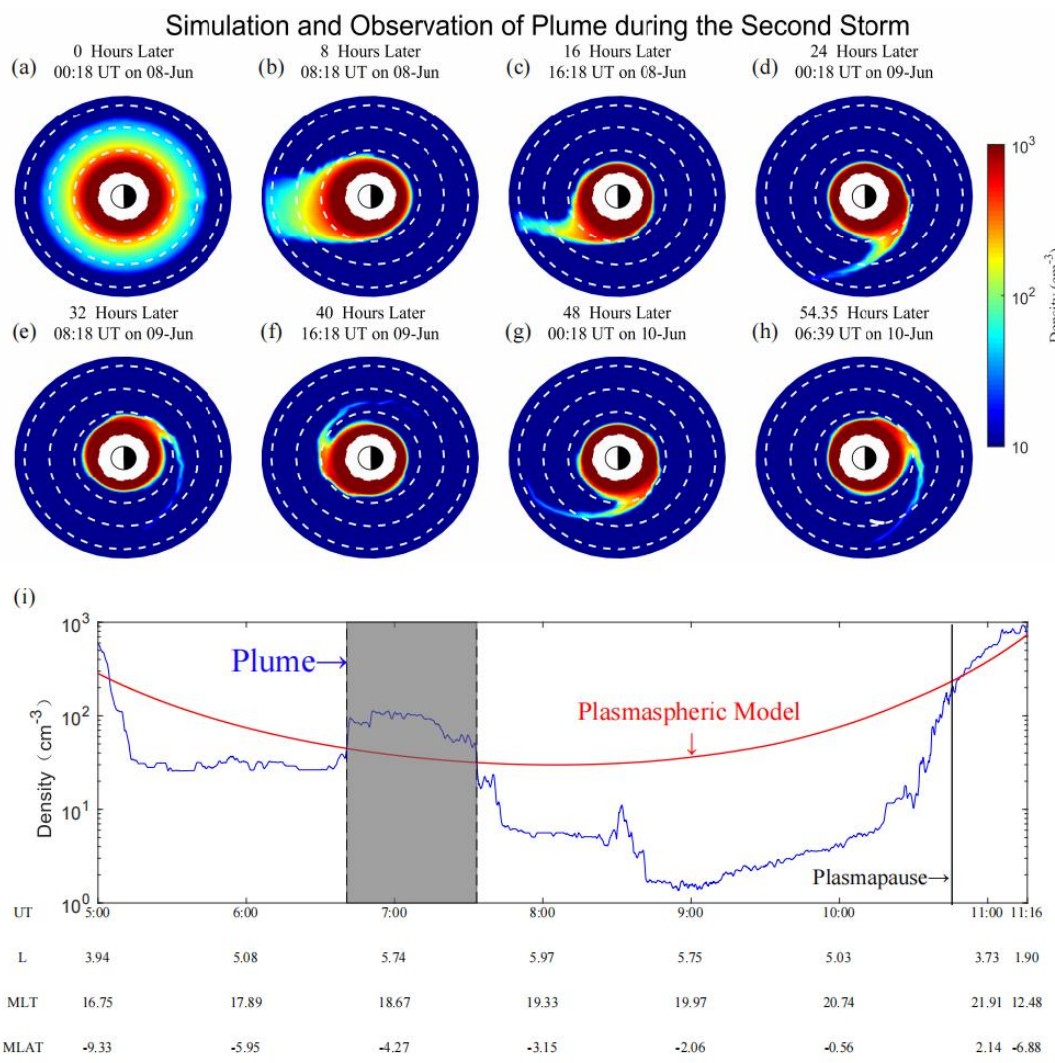

**Figure 10.** (a-h) Equatorial plots of the plasmasphere and plume obtained by simulation in the time interval of 08-10 June 2015. The white dotted circles, from inside out, indicate L shell values of 4, 6, and 8. The simulation duration and the corresponding actual duration are represented in the title. The solid white line in Figure 10h represents the orbit arc in which the VAP-A observed the plasmaspheric plume from 06:39 UT to 07:31 UT on 10 June. (i) The plasma density detected by VAP-A from 05:00 UT to 11:16 UT on 10 June 2015. The gray shadow indicates the plasmaspheric plume detected by VAP-A.






**Table 1.** The number of events corresponding to different intervals after the minimum Dst.

| Interval (Days) | 1 | 2 | 3 | 4 | 5 | 6 | 7 | >7 |
|---|---|---|---|---|---|---|---|---|
| Number of orbits | 66 | 54 | 51 | 54 | 42 | 40 | 90 | 26 |
| Number of plume events | 93 | 78 | 74 | 77 | 55 | 67 | 116 | 26 |





**Table 2.** The widths of the simulated plume at different L shells during the above two geomagnetic storms.

| Simulation results | The first storm (-39 nT) (18:12 UT on 20 May 2017) | | | The second storm (-74 nT) (06:39 UT on 10 June 2015) | | |
|---|---|---|---|---|---|---|
| L shell | 5.0 | 5.5 | 6.0 | 5.0 | 5.5 | 6.0 |
| MLT at entrance of the plume | 15.93 | 15.80 | 15.47 | 23.00 | 21.73 | 20.93 |
| MLT at exit of the plume | 18.53 | 17.47 | 16.53 | 23.47 | 22.13 | 21.07 |
| ΔMLT | 2.60 | 1.67 | 1.06 | 0.47 | 0.40 | 0.13 |
| The cartesian distance/$R_E$ | 3.34 | 2.38 | 1.67 | 0.61 | 0.58 | 0.21 |