# Peer review of "Width of Plasmaspheric Plumes Related to the Level of Geomagnetic Storm Intensity"

_Annales Geophysicae, 2022_

## Referee Comment (RC1)

Review on « Width of Plasmaspheric Plumes Related to the Intensity of Geomagnetic Storm » by Yang et al.

In their paper, Yang et al. address the question of the width plasmaspheric plumes and how it correlates with the intensity of the geomagnetic storm. Previous work (Borovski et al, 2008) already showed that there are no correlation (confirmed by this work when looking at the whole event), but this work showed that a negative linear correlation can be found when focusing only on the recovery phase of the storm.
To reach this conclusion, this paper used spacecraft data from the Van Allen mission (ranging from 2012 to 2019), used different definition of the plasmaspheric plume width and excluded extreme events to create a set of events in order to perform statistic analysis. It also use simulation data to explain its results, explaining in particular the inverse correlation by a stronger erosion of the plasmasphere during the initial and mains phase of the geomagnetic storm, resulting in narrower plumes in the recovery phase.

The results of this paper are clear and well explained. The use of simulation offers a nice development of the observation results, offering a new result of use for the community working on plasmaspheric plumes. I would recommend this paper for immediate publication if not for minor comments listed below, my main issue being some lack of clarity when manipulating the statistic tools.

**Main comments:**

l. 121 —> the set of data used for the Spearman (and Pearson) correlation has a strong constraint: it needs set of data with the same number of data. This first point is already unclear, as you don't say what is N (the number of data point). And then, given the constraint, it is also important that you develop clearly later how you define those set of data so that you can apply these methods.

In Fig 3 (and the text associated), you explain quite clearly how you select your events, and why you make those choices. However, it is interesting to see that purple point events (extreme events) always match quite well with orange points event (not in the recovery phase). A sentence or two about this and the possible explanation of this correlation would be interesting in the discussion.

l.229 —> The F, G and $V_S$ functions are not detailed later. You need more precisions. If this is too long for the core of the paper, maybe in annex.

**Minor comments:**

l.92: MLT (Magnetic Local Time, I guess) is not defined before use.

l.209: I'd prefer if you give the meaning of PTP, even if this is related to the quoted works.

l.253: typo: EIM —> $E_{IM}$

l.340: probably a typo (.javascript:void(0))

l.363: I think you can merge the financial support with the acknowledgment

l.490. Maybe detail the different units used in abscissa of the plot.

---

## Author Comment (AC2)

In their paper, Yang et al. address the question of the width plasmaspheric plumes and how it correlates with the intensity of the geomagnetic storm. Previous work (Borovsky et al, 2008) already showed that there are no correlation (confirmed by this work when looking at the whole event), but this work showed that a negative linear correlation can be found when focusing only on the recovery phase of the storm.

To reach this conclusion, this paper used spacecraft data from the Van Allen mission (ranging from 2012 to 2019), used different definition of the plasmaspheric plume width and excluded extreme events to create a set of events in order to perform statistic analysis. It also use simulation data to explain its results, explaining in particular the inverse correlation by a stronger erosion of the plasmasphere during the initial and mains phase of the geomagnetic storm, resulting in narrower plumes in the recovery phase.

The results of this paper are clear and well explained. The use of simulation offers a nice development of the observation results, offering a new result of use for the community working on plasmaspheric plumes. I would recommend this paper for immediate publication if not for minor comments listed below, my main issue being some lack of clarity when manipulating the statistic tools.

**Main comments:**

L. 121 —> the set of data used for the Spearman (and Pearson) correlation has a strong constraint:
it needs set of data with the same number of data. This first point is already unclear, as you don't say what is N (the number of data point). And then, given the constraint, it is also important that you develop clearly later how you define those set of data so that you can apply these methods.
In Fig 3 (and the text associated), you explain quite clearly how you select your events, and why you make those choices. However, it is interesting to see that purple point events (extreme events) always match quite well with orange points event (not in the recovery phase). A sentence or two about this and the possible explanation of this correlation would be interesting in the discussion.
**Answer:**
Thanks for your suggestion. We calculated the Pearson correlation between the average plume width in every 5 nT interval and the median Dst in the corresponding interval. Obviously, they have the same data length with N equals 15.
We notice the interesting phenomenon about the orange points and purple points event. For statistical results, this paper only analyzes the plume events with Dst value ranging from -90 nT to -15 nT during the recovery phase. The analysis of plume events in too weak magnetic storms will be our further work. And there are few plume events from -90 nT to -209 nT in the range of excessively negative Dst values (less than 12% plume events in the range of 57% Dst values), so analyzing this portion of the plume events would bring about a chance result. Therefore, we exclude the events

with excessively negative Dst values (less than -90 nT).

More information about N is supplemented in the new manuscript:

**On Lines 120-123:**

'As described above, we use the averaged width of the detected plume in steps of 5 nT (Dst index) to represent the plume width in the corresponding Dst range. N is equal to 15 if the geomagnetic activity levels from Dst~-15 nT Dst~-90 nT are considered in the study.'

The discussion about your suggestion is supplemented in the Discussion and Conclusion part of the new manuscript:

**On Lines 158-162:**

'This paper analyzes the plume events with Dst value ranging from -90 nT to -15 nT during the recovery phase. In the future study, we will use other geomagnetic activity indices to analyze the relationship between plume width and magnetic storm intensity, such as Kp, and AE, thus studying the correlation between the plume width and substorm activity.'

L.229 —> The F, G and VS functions are not detailed later. You need more precisions. If this is too long for the core of the paper, maybe in annex.

**Answer:**

Thanks for your suggestion. The specific expressions and modulation effect on SAPS of $F(r, \varphi)$, $G(\varphi)$ and $V_S(t)$ are supplemented in the revised manuscript:

**On Lines 234-243:**

'The detail formulas of functions are as following:

[1]. The function $F(r, \varphi)$ treats the SAPS flow channel as a potential drop centered at radius $R_s$:

$$F(r, \varphi) = \frac{1}{2} + \frac{1}{\pi}\tan^{-1}\left[\frac{2}{\alpha}\{r - R_s(\varphi)\}\right] \qquad (5)$$

where $R_s$, $\alpha$ are represented as:

$$R_s(\varphi) = R_0\left[\frac{1 + \beta}{1 + \beta\cos(\varphi - \pi)}\right]^\kappa \qquad (6)$$

with $\beta$=0.97 and $\kappa$=0.14. And $R_0$:

$$R_0/R_E = 4.4 - 0.6(K_P - 5) \qquad (7)$$

here $R_E$ is the radius of the Earth, 6380 km.

$\alpha$ is expressed as:

$$\alpha = 0.15 + (2.55 - 0.27K_P)\left[1 + \cos\left(\varphi - \frac{7\pi}{12}\right)\right] \qquad (8)$$

[2]. Based on the magnitude of the SAPS potential drop decreases eastward across the nightside, azimuthal modulation of SAPS magnitude $G(\varphi)$ is set to:

$$G(\varphi) = \sum_{m=0}^{2}\{A_m\cos[m(\varphi - \varphi_0)] + B_m\sin[m(\varphi - \varphi_0)]\} \qquad (9)$$

[3]. The $V_S(t)$ function describes the time regulation of SAPS:

$$V_S = (0.75kV)K_p^2 \qquad (10)$$

'

**Minor comments:**

L.92: MLT (Magnetic Local Time, I guess) is not defined before use.

**Answer:**

Thanks for your suggestion. The definition of MLT has been added to the first mentioned position of MLT in the revised manuscript:

**On Lines 93-94:**

'Second, the width is considered the difference between the magnetic local times (MLTs) of the two plume ends ($\Delta$MLT).'

L.209: I'd prefer if you give the meaning of PTP, even if this is related to the quoted works.

**Answer:**

Thanks for your suggestion. The meaning of PTP and the difference between PTP and our particles simulation have been added in the revised manuscript:

**On Lines 210-212:**

'In this study, this process differs from the plasmapause test particle (PTP) simulation which only provides the evolution of the plasmapause and plasmaspheric plume boundaries in Goldstein et al. (2004; 2014a; 2014b);'

L.253: typo: EIM $\longrightarrow$ $E_{IM}$

**Answer:**

Thanks for your reminder. This spelling mistake has been corrected in the revised manuscript:

**On Lines 265-266:**

'..., the maximum $E_{IM}$ index reached 0.4684 mV/m.'

L.340: probably a typo (.javascript:void(0))

**Answer:**

Thanks for your reminder. This mistake has been corrected in the revised manuscript:

**On Line 353:**

'..., width during the recovery phase of geomagnetic storms.'

L.363: I think you can merge the financial support with the acknowledgment

**Answer:**

Thanks for your suggestion. The acknowledgement and the financial support have been merged in the revised manuscript:

**On Lines 375-380:**

'**Acknowledgments and financial support.**

This research is supported by the National Natural Science Foundation of China (Nos. 42064009). The data of EMFISIS onboard Van Allen Probe are from (http://emfifisis.physics.uiowa.edu/Flight/. The Dst and KP data are provided by OMNI at http://cdaweb.gsfc.nasa.gov. This research has been supported by the

National Natural Science Foundation of China (grant nos. 42064009).'

L.490. Maybe detail the different units used in abscissa of the plot.
**Answer:**
Thanks for your suggestion. It is obviously more standardized for the paper after indicating the different units, so the units have appear in the Figure 1, Figure 8i and Figure 10i in revised manuscript:

**Figure 1.**

[Figure]

**Figure 8.**

[Figure]

**Figure 10.**

[Figure]

---

## Author Comment (AC3)

This paper use upper hybrid frequency inferred electron density obtained from Van Allen Probes measurements together with test particle simulations to statistically study the storm dependence of plume width. The paper is clear and well written. The results are important for better understanding the evolution of cold plasma, especially the plume region, during storms, which is critical for wave and particle dynamics in/near the core region of the outer radiation belt. The reviewer suggest to publish the paper after addressing the following minor comment.

1. It is not clear how storms are selected in this paper. Some storms even have a minimum Dst of close to 0 nT (Figure 3), which seem to be very small ones and are normally not considered as storms. Please explain more on how storm events are selected.

**Answer:**

Thanks for your suggestion. In the paper, we mainly focus on the plume events in the recovery phase of storm. In fact, Figure 3 shows the process of deleting events. All the potential plume events observed by the Van Allen Probe A from 2013 to 2018 are listed in Figure 3. The orange solid dots indicate the events that are not in the recovery phase, which are beyond 5 days after the minimum Dst. The purple solid dots indicate plume events with corresponding Dst values less than -70 nT or greater than -15 nT. The blue and red solid dots represent the tracks with $\triangle$ L less than 0.2 and Cartesian distances greater than 3.5 $R_E$, respectively. The gray and black solid dots indicate the retained plume events with corresponding Dst ranges from -90 nT to -70 nT and from -70 nT to -15 nT, respectively.

After filtering, there are 145 retained plume events in the Dst range from -70 nT to -15 nT, and 165 retained plume events in the Dst range from -90 nT to -15 nT.

The detailed description can be found **on lines 131-164** in the revised version of manuscript.

2. In addition to the average plume width dependence on storm intensities, there is also a clear dependence of the upper limit of plume width on storm intensities. This seems to be an interesting feature as well. The plume width at each orbit may depend on MLT, L, and storm phase. However, it tend to show a stable trend that how wide it can be during a storm during multiple measurement.

**Answer:**

Thanks for your suggestion. As your suggestion, we analyze the upper limit of plume width with Dst value ranging from -90 nT to -15 nT and form -70 nT to -15 nT during the recovery phase:

[Figure]

Obviously, similar to the results of the average plume width analysis, the upper limit of plume width is also negatively linear correlated with Dst. This interesting situation will be detailedly analyzed in further work.

3. Just to mention that Van Allen Probe is mostly short as RBSP instead of VAP, although it is optional to the authors.

**Answer:**

Thanks for your reminder. The VAP satellites, which are also called Radiation Belt Storm Probes, were launched on 30 August 2012. In this paper, we research the data set of VAP-A.

4. Line 69, date 09 October 2016 is not consistent with Figure 1 caption.

**Answer:**

Thanks for your reminder. This mistake has been corrected in the revised manuscript:

**On Lines 78-79:**

'The blue curve indicates the observed electron density from 08:00 UT to 16:36 UT on 19 October 2016. '

5. Line 112, it is not necessary to keep so many digits after the point. 2 or 3 digits should be good enough.

**Answer:**

Thanks for your suggestion. According to your suggestion, only three decimal places

are reserved in revised article and figures.

For examples:

**On Lines 111-114:**

'..., which indicates the linear relevance between the averaged Cartesian distance of plasmaspheric plumes (indicated by red asterisks) and the corresponding Dst value, is only -0.017. The P-value (marked as P), which is generally adopted to express the reliability of their linear correlation, reaches 0.935.'

**On Lines 175-177:**

'As presented in Figure 5a, the Pearson correlation coefficient between the averaged Cartesian distance of plumes and the Dst value reaches 0.619. The value of the P-value is 0.042, ...'

6. Lines 140 and 141, typo 'tthe' and 'ro'

**Answer:**

Thanks for your reminder. These spelling mistakes have been corrected in the revised manuscript:

**On Lines 143-144:**

'..., and 88% of plume events (333 events) correspond the minimum Dst value of the intensest geomagnetic storm to Dst values ranging from -90 nT to -15 nT.

---

## Author Comment (AC4)

The paper "Width of Plasmaspheric Plumes Related to the Intensity of Geomagnetic Storm" investigates the relationship between the minimum value of dst indice and the plasmaspheric plume width. The dst indice is assumed to be connected to geomagnetic storm intensity. The analysis relies on Van Allen Probes data and it is illustrated with simulations of two plume events. The paper is logically organized, the figures are generally correctly described and the reasoning is clearly explained. However, I have several concerns about the methodology that I would recommend to address before publication.

The Dst is the single proxy for geomagnetic storm intensity considered in the study. Looking back at Gonzales 1994, based on magnetic storms, only min(Dst) lower than -30nT are related to magnetic storms (weak:[-50,-30] and moderate [-50,-100] nT). Intense magnetic storms are not part of this study and conversely the large number of events with min(Dst) larger than -30nT might not be considered as magnetic storms. Your work seems to analyze the relationship between plume width and Dst dip (rather than magnetic storm intensity). This point needs to be clarified (for example, you could either replace the wording "magnetic storm intensity" or exclude the >-30nT events from the study).

**Answer:**
Thanks for your suggestion. Dst value reflects the level of magnetic storm intensity. According to your suggestion, we replace the wording "magnetic storm intensity" with "level magnetic storm intensity". Of course, combining the relationship between plume width and Kp or AE indices can better explain the relationship between plume width and "magnetic storm intensity", which will be done in further work.

The discussion about your suggestion has supplemented in the "Discussion and Conclusion" part of revised manuscript:

**On Lines 358-362:**
'This paper analyzes the plume events with Dst value ranging from -90 nT to -15 nT during the recovery phase. In the future study, we will use other geomagnetic activity indices to analyze the relationship between plume width and magnetic storm intensity, such as Kp, and AE, thus studying the correlation between the plume width and substorm activity.'

You claim that you find a relationship a dependence between the plume width only when excluding the events with most negative min(Dst) values. To my opinion it is not fully proven, as you also excluded events that are too large (in MLT and/or R). I would suggest to also try to find a relationship between the plume width and all the dst values when the large plumes are excluded.

**Answer:**
Thanks for your suggestions. We do not claim that the correlation value arose only after excluding plume events with most negative min (Dst) values. There are few plume events from -90 nT to -209 nT in the range of excessively negative Dst values (less than 12% plume events in the range of 57% Dst values), so analyzing this portion of the plume events would have resulted in a chance result. Therefore, we

exclude the events with excessively negative Dst values (less than -90 nT).

I have some concern about your statistical analysis (L204). I do not see any very significant linear relationship in Figure 6, as stated L204. Is it a problem of data distribution, of Spearman correlation understanding, or something else?
**Answer:**
Thanks for your reminder. We calculate the Pearson coefficient between the P to indicate the size of their linear correlation, and calculate P-value (I'm sorry that P-value was wrongly written as Spearman correlation coefficient in the manuscript. This terrible error has been corrected in the new manuscript.) to test the feasibility of their non correlation. These calculations are realized by the corrcoef function in the MATLAB software.
In order to avoid misunderstanding, the wrong sentence has been changed to:
**On Lines 205-206:**
'The minimal P-value (~2.890×10-10 and ~4.300 ×10-7, respectively) shows that the linear relationship between them is very reliability.'

**Minor comments:**

It is striking from your simulation figures that the plume width is smaller at larger R values. How does this affect your analysis?
In the first paragraph a reference to a paper of early explanation/observation of the plasmaspheric plume is missing.
**Answer:**
Thanks for your reminder. Limited by the method of measuring plume width, we analyze the average plume width instead of each plume width to statistically reduce the impact of different R on plume width. At the same time, it also reflects the importance of our simulation, which can show a more comprehensive process of plume evolution to verify our statistical conclusions.

L63 (1): this is true for outbound crossing only. I guess you identify the plasmapause during inbounds as well.
**Answer:**
Thanks for your reminder. A complication to the measurement of plume widths is that VAP-A satellite always work anticlockwise, at same time, the plume is still rotating clockwise (mostly) or anticlockwise. This effect can lead the observed plume width wider or narrower than actual one. In a statistical sense, since the entry edge of the plume has similar behavior to the exit edge of the plume, the average width of many plumes is fairly insensitive to such effects (Borovsky et al., 2008).

L83-84: Need some clarification. How did you proceed? Did you search the plumes for the whole rbsp dataset or are you looking for plumes only after Dst minima?
**Answer:**

Thanks for your reminder. Referring to Halford et al. (2010) and Wang et al. (2016), the onset time of storm is defined as the time when the Dst index slope becomes negative and keeps relatively negative till the minimum of Dst index in our study. In this paper, we search the plumes for the VAP-A dateset (from 2013 to 2018), and think the intensity of magnetic storm is represented by the minimum Dst value from the onset time of nearest storm to the time when plume is identified.

L134: how do you define the main phase? How many plumes are not related to any storm?

**Answer:**

Thanks for your reminder. Referring to Halford et al. (2010), the main phase is defined as the period from onset of the storm until the Dst reaches minimum value.

We believe that the plume cannot be generated by a too weak magnetic storm, so the plume events with less than -15 nT are considered to be the residual part of the plumes generated by earlier magnetic storm and are finally excluded.

So, the definition of main phase has been added in the revised manuscript:

**On Lines 134-137:**

'Similar to the standards described in Engebretson et al. (2008), Halford et al. (2010) and Bortnik et al. (2008), we define the main phase is the period from onset of the storm until the Dst reaches its minimum value, and the end of the recovery phase is defined as the fifth day after the main phase finishes.'

L209: describe in few words how the two processes differ.

**Answer:**

Thanks for your suggestion. By calculating the drift paths of a great quantity of test plasmaspheric particles, the simulation not only provides the evolution of the plasmapause and plasmaspheric plume boundaries, which is similar to the plasmapause test particle (PTP) simulation provided by Goldstein et al. (2004; 2014a; 2014b), but also reveals the evolution of equatorial density in both the plasmasphere and plasmaspheric plume.

So, the sentences have been revised in the new manuscript:

**On Lines 210-212:**

'In this study, this process differs from the plasmapause test particle (PTP) simulation which only provides the evolution of the plasmapause and plasmaspheric plume boundaries in Goldstein et al. (2004; 2014a; 2014b);'

L223: one can suppose that Maynard and Chen discovered the solar wind electric field.

Is that correct?

**Answer:**

Thanks for your reminder. I'm sorry we misunderstood you by using references incorrectly. Maynard and Chen (1975) give a formula about the correlation between solar wind electric field and model of convection electric potential ($\Phi_{VS}$). In the new manuscript, we adjust the position of this reference to avoid misunderstanding:

**On Line 223:**

'the convection electric potential, which is expressed as (Maynard and Chen, 1975): ...'

L223: "IMF" azimuthal angle

**Answer:**

Thanks for your reminder. $\varphi$ indicates the azimuthal angle, and the relationship with MLT is MLT = $12(\varphi/\pi + 1)$.

The supplement about azimuthal angle has been added in the revised manuscript:

**On Lines 226-227:**

'Here, the ESW is the solar wind electric field. $\varphi$ indicates the azimuthal angle, where MLT = $12(\varphi/\pi + 1)$.'